# Position: Trustworthy AI Suffers from Invariance Conflicts and Causality is the Solution

Ruta Binkyte [* 1]  Ivaxi Sheth [* 1]  Zhijing Jin [2 3 4]  Mohammad Havaei [5]  Bernhard Schölkopf [2 6]  Mario Fritz [1]

## Abstract

As artificial intelligence (AI), including machine learning (ML) models and foundation models (FMs), are increasingly deployed in high-stakes domains, ensuring their trustworthiness has become a central challenge. However, the core trustworthy AI objectives, such as fairness, robustness, privacy, and explainability, are hard to achieve simultaneously, especially while preserving utility. **This position paper argues that causality is necessary to understand and balance trade-offs in performance and multiple objectives of trustworthy AI.** We ground our arguments in re-interpreting trustworthy AI trade-offs as incompatible invariance requirements under different changes to the data-generating process. We then illustrate this argument through case-study analyses from the literature and a stylized synthetic-data simulation, showing that causality provides a unifying framework for understanding how trade-offs in trustworthy AI arise and how they can be softened or resolved through selective invariance. This perspective applies to both classical ML models and large-scale FMs. Finally, we outline open challenges and opportunities for using causality to build both trustworthy and high-performing AI.

## 1. Introduction

Machine learning (ML) models have driven remarkable advances in natural language processing, computer vision, and decision-making, enabling large-scale deployment in domains such as healthcare, finance, education, and social media. More recently, foundation models (FMs), including large language models (LLMs) and vision language models (VLMs), have demonstrated unprecedented generality across tasks (Achiam et al., 2023; Team et al., 2023; Radford et al., 2023). Given their influence, ensuring ethical and trustworthy ML and FM systems has become a global priority. Many international regulations and frameworks (European Commission, 2021) seek to establish guidelines for AI that would avoid harmful social impact. In this work, we focus on four trustworthy dimensions - fairness, explainability, robustness, and privacy protection - as well as their relationship with model performance.

**Trade-offs in trustworthy AI.** The objectives in trustworthy AI[1] are rarely independent. Improving one aspect often comes at the expense of another, or model performance. For example, **privacy** noise added to protect data reduces model accuracy (Xu et al., 2017; Carvalho et al., 2023). Similarly, achieving **fairness** frequently requires sacrificing predictive performance or resolving conflicts between competing fairness notions, such as demographic parity and equalized odds (Friedler et al., 2021; Kim et al., 2020). Performance is also frequently traded off against **explainability**, as complex deep models excel in accuracy but are not comprehensible by humans (Crook et al., 2023). Fairness and privacy are often mutually reinforcing, but can conflict when the loss in performance disproportionately hurts sensitive groups (Pujol et al., 2020). Similarly, overfitting on spurious signals in the data gives accuracy at the cost of **robustness** (Tsipras et al., 2019). However, much of the machine learning research and development has historically prioritized improving predictive accuracy and performance or treated these trade-offs as empirical side effects.

**Causality and trustworthy AI.** Unlike correlation-based approaches, causal models represent the mechanisms that produce data, enabling selective invariance: they allow models to be invariant to spurious pathways while remaining sensitive to stable, causally meaningful signals. Causal methods have already shown promise

---
[*]Equal contribution [1]CISPA Helmholtz Center for Information Security [2]Max Planck Institute for Intelligent Systems, Tübingen [3]EuroSafeAI [4]Jinesis Lab, University of Toronto & Vector Institute [5]Google Research [6]ELLIS Institute Tübingen. Correspondence to: Ruta Binkyte <ruta.binkyte@gmail.com>.

*Proceedings of the 43rd International Conference on Machine Learning*, Seoul, South Korea. PMLR 306, 2026. Copyright 2026 by the author(s).

---
[1]Throughout the paper, we use *AI* as an umbrella term for learning-based models, including both classical machine learning models and foundation models.

in auditing and mitigating unfairness (Kim et al., 2021; Kilbertus et al., 2017; Loftus et al., 2018) and in improving robustness under distribution shift (Schölkopf et al., 2021). In addition, recent theoretical results show that under certain conditions robustness implies causality (Richens & Everitt, 2024). While the connection between causality and privacy is still less explored, early work indicates that causal structure can improve privacy-utility trade-offs (Tschantz et al., 2020; Tople et al., 2020; Binkyte et al., 2024). Finally, explainability is inherently aligned with causality, as explanations are naturally expressed in terms of interventions and counterfactuals. However, despite recognition of causal solutions to individual challenges of trustworthy AI (Liu et al., 2023; Rawal et al., 2025; Ganguly et al., 2023), its role in reconciling the trade-offs between multiple goals and model performance remains underexplored.

**Position.** In this paper, *we argue that the trade-offs in trustworthy AI are not incidental, but arise from incompatible invariance requirements imposed by individual trustworthy AI objectives*. Robustness demands stability under distributional shifts; fairness demands stability when protected attributes or group membership change; privacy demands stability when individual data points are added or removed; and explainability demands stable attribution of changes under feasible actions. *At their core, trustworthy AI objectives can be reformulated as demands that a model's behavior remains stable under changes.* When different trust objectives demand stability under different and potentially conflicting sets of changes, trade-offs become unavoidable. For instance, accuracy under the observational distribution may rely on correlations that are unstable under interventions on the sensitive attributes or environmental variables, while fairness and robustness explicitly demand invariance to such interventions. Based on the above, **we position causality as a unifying framework for understanding how trade-offs in trustworthy AI arise, and how they can be softened or resolved.**

We support the position with a simulated example and show how causal knowledge can yield improvement across all four trust dimensions simultaneously without damaging accuracy.

**When causality is necessary?** Not all invariance requirements immediately necessitate causal reasoning. If a single trust objective is considered in isolation and the admissible changes are limited, purely statistical methods may suffice to enforce invariance. For example, a classifier can be trained to satisfy independence between the sensitive attribute and the outcome on a fixed dataset, or to be robust to a specific, observed form of covariate shift, without any explicit causal model. However, this breaks down as

soon as additional objectives or accuracy constraints are imposed. When multiple invariances must hold simultaneously, it becomes necessary to distinguish which dependencies are spurious and which reflect stable mechanisms. This distinction cannot, in general, be made from observational data alone. Importantly, this necessity is agnostic to model class. It applies equally to classical machine learning models and to large-scale foundation models. The distinction lies not in whether causal reasoning is needed, but in how causal assumptions are encoded and operationalized at scale.

**How causality resolves the trade-offs in trustworthy AI?** Causal reasoning is therefore required, not merely to enforce a single constraint, but to reason about which invariances can coexist, which must conflict, and how trade-offs can be selectively navigated. More precisely, causality allows:

(i) *Selective invariance through structural constraints.* The causal graph clarifies which pathways are subject to invariance requirements, allowing constraints to be enforced selectively rather than uniformly. This makes it possible to suppress normatively unacceptable effects, such as direct or proxy discrimination, while preserving causally justified signal. The same structure can also show when distinct requirements share a pathway, so that one intervention can serve several objectives at once.

(ii) *Shift focus from observational accuracy to interventional validity.* Causal reasoning distinguishes correlations that merely predict outcomes from causal mechanisms and reduces overfitting to spurious correlations. This resolves the tension between predictive accuracy and trustworthy objectives by shifting the optimization target from performance under the observational distribution to correctness under interventions.

**Paper map and contributions.** We reinterpret trustworthy AI literature to illustrate how trustworthy AI objectives translate into competing invariance requirements under interventions. We then show how causal structure enables principled resolution of these conflicts and discuss how causality can be encoded in both classical ML systems and foundation models. We discuss the conceptual and practical limitations of resolving trustworthy AI trade-offs using a causal approach. Finally, we draw future directions and call for actions necessary to overcome these limitations.

Our contributions are the following: 1.) We reinterpret major trade-offs in trustworthy AI through a conflicting invariance requirements lens; 2.) We formulate when causal reasoning is necessary and how it resolves or softens trade-offs in trustworthy AI; 3) We distinguish explicit and implicit causality and discuss how causal assumptions may be applied explicitly or implicitly in modern large-scale systems. 4) We provide an illustrative simulation showing how ex-

plicit or implicit causal models help to improve trustworthy AI trade-offs.

## 2. Background and Preliminaries

We introduce the core dimensions of trustworthy AI, emphasizing formal characterizations that are relevant for understanding trade-offs and motivating causal reasoning.

### 2.1. Interventions and Invariance

Let $\mathcal{D}$ denote the observational data distribution over $(X, Y)$, and let $\{\mathcal{D}^I\}_{I \in \mathcal{I}}$ denotes a family of distributions obtained by applying a set of admissible changes $I$ to the data-generating process. These changes may correspond to modifications of inputs, environments, data collection procedures, population characteristics, or other aspects of the data-generating process. A model is said to satisfy an invariance requirement with respect to a set of admissible changes $\mathcal{I}$ if its behavior or performance remains stable across the corresponding distributions $\{\mathcal{D}^I\}_{I \in \mathcal{I}}$.

### 2.2. Trustworthy ML Objectives

We now instantiate the invariance principle defined above for common trust dimensions. Consider a supervised learning setting with observed features $X \in \mathcal{X}$, a target variable $Y \in \mathcal{Y}$, and a learned predictor $f : \mathcal{X} \to \hat{\mathcal{Y}}$. Each trust objective corresponds to requiring $f$ to satisfy an invariance requirement with respect to a specific class of admissible changes. These invariance requirements differ in *what* is intervened on and *when* invariance is enforced: fairness and robustness target test-time interventions on protected attributes and environments, respectively; privacy targets training-time interventions on the dataset, and explainability concerns, selective sensitivity to feature-level interventions at inference.

**Fairness (invariance to protected attributes).** Let $A \in \mathcal{A}$ denote a sensitive or protected attribute. Fairness requires that the behavior of $f$ remain *invariant* under admissible changes to $A$, either in distribution (e.g., Demographic parity) or conditional on $Y$ (e.g., Equalized odds). In foundation models, the same invariance requirement instantiates with $X$ as the prompt or context and $Y$ as the completion: a fairness-invariant model should produce semantically equivalent completions when a sensitive token in $X$ is substituted (e.g., "he"$\to$ "she" in a recommendation-letter prompt), while allowing legitimate, task-relevant differences.

**Privacy (invariance to data inclusion).** Let $D$ denote a training dataset and let $D'$ be a neighboring dataset that differs from $D$ in a single individual. Privacy requires

that the behavior of a randomized mechanism $M$ remain approximately *invariant* under the admissible change $D \to D'$. In other words, the output distribution of $M(D)$ should be stable with respect to the inclusion, removal, or modification of any single data point.

**Robustness (invariance to distribution shifts).** Let $\{\mathcal{D}_e\}_{e \in \mathcal{E}}$ denote a family of distributions indexed by environments $e$. Robustness requires that the behavior or performance of a model remain *invariant* across these environments. Formally, this is captured by bounded loss across the family $\{\mathcal{D}_e\}_{e \in \mathcal{E}}$, where each $e$ may correspond to a shift in population, measurement process, or data-generating mechanism.

**Explainability (selective invariance under changes).** Let $\mathcal{I}_{\text{rel}}$ and $\mathcal{I}_{\text{irr}}$ denote sets of admissible relevant and irrelevant changes to the input or context, respectively. Explainability concerns the ability to understand and justify a models predictions through how they respond to these changes. From the invariance perspective, $f(X)$ should be *invariant* under changes in $\mathcal{I}_{\text{irr}}$, and responsive under changes in $\mathcal{I}_{\text{rel}}$ in a predictable and meaningful way. This definition equally applies to modern interpretability methods.

### 2.3. Accuracy

Accuracy measures predictive performance under the observational distribution $\mathcal{D}$, typically via the expected loss $\mathbb{E}_{(X,Y) \sim \mathcal{D}}[\ell(f(X), Y)]$. In practice, accuracy often conflicts with invariance-based trust requirements, giving rise to the trade-offs studied in this work.

### 2.4. Interventional Accuracy

Interventional accuracy measures predictive performance under the interventional distribution $\mathcal{D}^\mathcal{I}$, and can be expressed as the expected loss $\mathbb{E}_{(X,Y) \sim \mathcal{D}^\mathcal{I}}[\ell(f(X), Y)]$. In contrast to observational accuracy, which rewards correlations that hold in the training environment, interventional accuracy prioritizes predictors whose performance is stable across the family of interventions.

### 2.5. Causality

Purely associational learning methods operate on the observational distribution $\mathcal{D}$ and are sufficient for answering queries of the form $P(Y \mid X)$. However, many trustworthy ML objectives require reasoning about how predictions would change under hypothetical or interventional modifications to the data-generating process. To formalize such changes, we adopt Pearls structural causal framework (Pearl, 2009).

Central to Pearl's framework, an SCM consists of a set of endogenous variables $V$, exogenous variables $U$, and structural equations $F$ that determine how each variable in $V$ is generated from its parents and noise. This structure is summarized by a causal graph $\mathcal{G} = (V, \mathcal{E})$, which is a directed acyclic graph (DAG) where $\mathcal{E}$ is a set of edges connecting the nodes $V$.

Given a causal graph $\mathcal{G}$, a *causal path* from a variable $X$ to a variable $Y$ is any directed sequence of edges in $\mathcal{G}$ that begins at $X$ and ends at $Y$. Causal paths characterize how the effect of one variable on another is transmitted through intermediate variables, and provide a decomposition of the total causal effect into distinct mechanistic components.

## 3. Trade-offs as Invariance Conflicts

In this section, we build on concrete examples and reinterpret well-known results from the literature to show that many trade-offs in trustworthy AI arise from fundamentally incompatible invariance requirements imposed by different objectives. We then highlight how causal reasoning or implicit causal data interventions provide a principled way to diagnose and soften or resolve these conflicts. In addition, we provide a motivating example and numerical simulation (see the Motivating Example 3.5) as a concrete illustration of the discussed ideas.

### 3.1. Fairness–Accuracy Trade-off

**Statistical origin of the trade-off.** Predictive accuracy under the observational distribution $\mathcal{D}$ often relies on correlations between features and sensitive attributes $A$. When these correlations are predictive, suppressing them to satisfy fairness constraints can reduce accuracy. This phenomenon is well documented across fairness-aware ML learning methods (Zliobaite, 2015; Zhao & Gordon, 2022). Similarly, in generative models, careless fairness interventions can backfire; for instance, efforts to enhance diversity have led to historically inaccurate outputs, as seen in critiques of Google's Gemini (Vincent, 2024).

**Invariance conflict.** Many fairness notions can be expressed as invariance requirements under interventions on $A$. For instance, Demographic parity can be interpreted as requiring invariance of the decision distribution $f(X)$ for all values of $A$. In contrast, maximizing accuracy under $\mathcal{D}$ encourages sensitivity to any predictive signal, including those downstream of $A$.

This creates a conflict between fairness and accuracy goals. Without distinguishing normatively unacceptable pathways from causally justified ones, these requirements are incompatible.

**Causal resolution.** Causal models distinguish between admissible and inadmissible causal pathways from $A$ to $Y$ (Chiappa, 2019). By explicitly modeling the data-generating process, causal reasoning enables selective invariance: suppressing only those paths deemed unfair (e.g., direct or proxy discrimination), while preserving legitimate causal effects (e.g., through explaining variables, such as gender dependent disease risks).

This logic applies across model classes. In classical ML, causal constraints operate on input features and representations. In foundation models, the same causal pathways are encoded implicitly in embeddings, attention patterns, or generation dynamics. Causal interventions, such as counterfactual data augmentation, causal disentanglement, or SCM-based regularization, can mitigate biased generative behavior while preserving task-relevant predictive structure (Zhou et al., 2023; Madhavan et al., 2023). Entity substitutions can selectively block unfair or spurious causal paths while preserving task-relevant information, illustrating how causal intervention softens the fairness-accuracy trade-off (Wang et al., 2023).

### 3.2. Privacy–Utility (and Attribution)

**Statistical origin of the trade-off.** Privacy-preserving mechanisms such as differential privacy reduce the influence of individual data points or sensitive attributes on model outputs by introducing privacy noise. While this limits information leakage, it also degrades utility by attenuating informative signals. In large-scale generative models, privacy risks additionally arise through memorization and indirect reproduction of personally identifiable information, further intensifying the tension between privacy and model usefulness (Xu et al., 2017).

**Invariance conflict.** Privacy requires invariance to interventions on private or individual-level variables, such that small changes to sensitive information do not substantially affect outputs. Utility, in contrast, requires sensitivity to variables and pathways that encode task-relevant information. In foundation models, this conflict extends to attribution: models must be insensitive to whether a specific individuals data is present in the training set, while remaining sensitive to generalizable patterns.

**Causal resolution.** Causal models mitigate the privacy–utility trade-off by reducing reliance on spurious correlations that lead to overfitting and memorization, which are primary drivers of re-identification and membership or attribute inference attacks. By aligning learning with stable causal mechanisms, such models are less susceptible to privacy attacks that exploit dataset-specific artifacts. In particular, Tople et al. (Tople et al., 2020) show that causal

models provide stronger defenses against membership inference attacks while requiring a smaller privacy budget $\epsilon$ under differential privacy, thereby achieving comparable privacy guarantees with less degradation in utility.

In addition, explicitly modeling how private attributes propagate through the system, causal structure makes it possible to suppress only privacy-sensitive effects while preserving non- sensitive causal relationships. We highlight this form of targeted causal obfuscation as a promising design direction that can lower the impact of data obfuscation on performance. Similarly, in foundation models, causal auditing can enable principled attribution by distinguishing data memorization from incidental stylistic similarity (Sharkey et al., 2024).

### 3.3. Robustness–Accuracy

**Statistical origin of the trade-off.** High predictive accuracy under a fixed training distribution often relies on shortcut correlations that are specific to that environment. When deployment conditions change, these correlations may break, leading to degraded performance. Methods that improve robustness by discouraging such shortcuts often reduce in-distribution accuracy.

**Invariance conflict.** Robustness requires invariance across a family of environments, which can be formalized as distributions related by interventions on latent or observed environmental variables. Accuracy under the observational distribution, however, rewards sensitivity to all predictive correlations, including those that are unstable under such interventions.

**Causal resolution.** Causal reasoning resolves this conflict by distinguishing invariant causal mechanisms from spurious associations. Features that are causal parents of the target remain predictive under interventions, while non-causal shortcuts do not. In this way, causal reasoning shifts focus to interventional accuracy and provides a principled way to prioritize stable features.

In foundation models, shortcut correlations are often encoded implicitly in learned representations. Causal invariance and regularization techniques can reduce reliance on unstable patterns while preserving task-relevant structure (Zhou et al., 2023). Moreover, recent theoretical results show that agents robust across environments must implicitly learn causal world models (Richens & Everitt, 2024), further highlighting the link between robustness and causality.

### 3.4. Explainability–Performance

**Statistical origin of the trade-off.** Highly expressive models often achieve strong performance by exploiting complex, distributed correlations that are difficult for humans to interpret. As a result, improvements in predictive capability frequently come at the cost of interpretability and explainability (London, 2019; Van der Veer et al., 2021).

**Invariance conflict.** Explainability requires stability of model behavior under interventions on semantically meaningful variables, enabling counterfactual reasoning about why a prediction was made. Performance-oriented models, in contrast, may rely on opaque correlations that change unpredictably under such interventions.

**Causal resolution.** Causal models address this tension by explicitly representing how inputs influence outputs through causal mechanisms, enabling counterfactual explanations and actionable recourse (Wachter et al., 2017; Koh et al., 2020; Sheth & Ebrahimi Kahou, 2023).

In foundation models, causal structure can be probed within internal components such as embeddings, attention heads, and logits. Methods for causal path analysis and intervention-based probing enable step-by-step explanations of generative behavior (Bagheri et al., 2024; Conmy et al., 2023). In addition, the causal approach in mechanistic interpretability provides a principled way to faithfully translate complex generative processes to human-understandable abstractions (Geiger et al., 2025).

### 3.5. Towards Multi-Objective Trade-Off Resolution

**Origins of multi-objective trade-offs.** The preceding sections show that many trade-offs between fairness, privacy, robustness, and accuracy or model performance arise when invariance constraints are enforced uniformly over all statistical dependencies. In addition, privacy mechanisms often worsen fairness for minority groups. When obfuscation is applied uniformly, the signal-to-noise ratio of underrepresented populations collapses first, because their causal pathways are already weakly supported by data. This creates a fairness-privacy trade-off. Similarly, data obfuscation obscures the relationships between predictive variables and the outcome, thus creating a privacy-explainability trade-off.

**Towards causal resolution.** We argue that the causal approach leads to the improvement of multi-objective trade-offs in trustworthy AI, or, in other words, allows to improve multiple dimensions simultaneously. First, by decomposing total effects into causal pathways, causal models allow invariance requirements to be targeted rather than global. For example, privacy can be enforced only on paths that transmit individual identifying information. Second, causal models are inherently explainable, robust, and require less privacy noise for preventing leakage of sensitive

information (Tople et al., 2020). As a result, they should better preserve global accuracy as well as accuracy for sensitive and underrepresented groups. Finally, in algorithmic recourse, causal knowledge allows specifying which interventions are feasible and admissible, and achieving fairness and robustness under counterfactual changes (Karimi et al., 2021). This makes causality a favorable design principle for future trustworthy AI systems and motivates systematic study of how these dimensions interact and can be reconciled through causal modeling.

## Motivating Example: Invariance Conflicts in Clinical Readmission Prediction

To make the invariance-conflict perspective concrete, we construct a structural causal model (SCM) based on the causal graph in Figure 1 and generate synthetic data from the distribution induced by its structural equations.

**Scenario.** The SCM represents a stylized 30-day clinical readmission setting in which a model is trained to predict whether a patient will be readmitted after discharge. The protected attribute $A$ denotes the age group. Age may be statistically associated with readmission, but not all effects of $A$ are admissible: laboratory measurements $X_1$ represent a legitimate clinical pathway, insurance type $Z$ is treated as an unfair proxy for $A$, and a direct effect $A \rightarrow Y$ represents inadmissible age-based influence.

Hospitals also differ in two ways. First, the hospital environment $E$ can affect readmission through genuine policy differences, represented by $E \rightarrow Y$. Second, it can induce environment-specific measurement artifacts $X_2$, such as hospital-dependent logging or imaging artifacts, which can be predictive without reflecting stable clinical risk. We therefore hold out one hospital environment as an unseen out-of-distribution (OOD) test environment.

**Invariance conflicts.** The SCM separates three distinct requirements. Fairness requires blocking the direct and proxy-mediated effects of age, $A \rightarrow Y$ and $A \rightarrow Z \rightarrow Y$, while preserving the legitimate clinical path $A \rightarrow X_1 \rightarrow Y$. Robustness requires ignoring the unstable artifact path through $X_2$, but not treating all hospital effects as spurious, since $E \rightarrow Y$ captures real policy differences. Privacy requires limiting the leakage of sensitive insurance information ($Z$) without destroying the useful clinical signal.

**Methods.** We compare five methods: statistical prediction with no intervention; a non-targeted privacy mechanism that adds Laplace noise to all features; a fairness-constrained method that drops $A$ and adds a penalty enforcing invariance of predictions to $A$; a partial implicit causal method using data-level interventions on the proxy $Z$ together with multi-environment training, and an explicit causal model that drops inadmissible and non-causal variables by graph-based feature selection.

**What the simulation illustrates.** Methods that enforce only a single global constraint achieve improvement at the expense of other dimensions. A statistical fairness intervention, lacking the graph, cannot separate the inadmissible path $A \rightarrow Z \rightarrow Y$ from the admissible $A \rightarrow X_1 \rightarrow Y$, so it attenuates both with a substantial accuracy and explainability drop and no gain on robustness. Adding privacy noise to all features costs accuracy but does not eliminate $Z$ leakage, since the adversary recovers $Z$ by exploiting population-level correlations with public features. However, it improves fairness as a by-product (and vice versa) because both objectives are addressed by reducing the model's reliance on $Z$. By contrast, the explicit Causal model directly targets and removes inadmissible pathways based on the underlying causal graph; it is best on fairness, privacy, and explainability by design. The Partial Implicit Causal method, which uses only weaker causal knowledge together with training across many environments, is second to the explicit causal model, and surpasses it in out-of-distribution environments. Observation that an implicit method can match or surpass the explicit on some axes is promising, since the full causal graph is rarely available in practice.

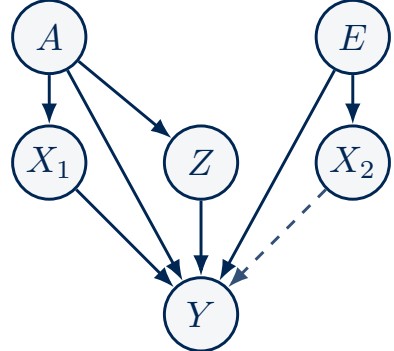

**Figure 1.** Causal graph for the readmission example. $A$: age group; $X_1$: labs; $Z$: insurance type (proxy of $A$); $E$: hospital; $X_2$: artifact; $Y$: readmission. Dashed edge: spurious association. Note: The causal model has access to the causal graph which qualitatively describes the relationships between the variables. However, it does not have access to full SCM with quantitative parameters of structural equations.

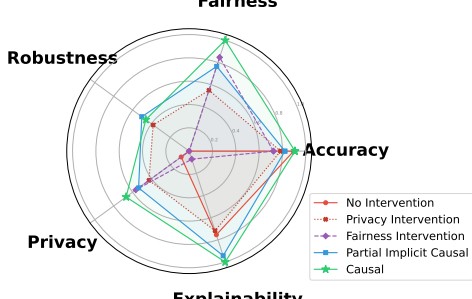

**Figure 2.** Six methods compared across accuracy (AUC-ROC), fairness (low prediction sensitivity to the proxy $Z$), robustness (Brier score on a held-out hospital), privacy (resistance to an attribute-inference attack on $Z$), and explainability (share of prediction variance from causally justified features). All metrics are normalized and oriented so that higher is better on every axis.

**Interpretation note.** All results come from a single SCM chosen to give a clear instance of the trade-offs and synergies in trustworthy AI the paper discusses. The simulation serves as a stylized proof of concept example supporting the position. Quantifying the generality of the results is the task for future work. Technical implementation is described in Appendix A. Limitations are further discussed in subsection A.5 of the Appendix. The simulation and analysis code is accessible at GitHub repository causal-invariance-conflicts-ai.

# 4. Integrating Causality into ML and Foundation Models

In this section, we discuss how causality can be encoded in learning systems, spanning both classical ML models and modern foundation models (FMs). We focus on the high-level principles relevant to the feasibility of the causal approach. For a detailed overview of the methods see (Binkyte et al., 2025).

## 4.1. Defining Explicit And Implicit Causal Integration

Causal assumptions can be incorporated into learning systems either explicitly, through structural representations, or implicitly, through inductive biases and training regimes that encourage causal behavior (Rawal et al., 2025). Based on this distinction, we define *Explicit* and *Implicit* approaches to causality in ML and FMs.

**Explicit causal integration.** Explicit approaches represent causal structure directly, typically via structural causal models (SCMs) or causal graphs, and learning objectives are defined to satisfy causal properties under specified interventions.

Formally, given a causal graph $G$, a model $f_\theta$ is trained to satisfy invariance or sensitivity constraints derived from $G$.

The primary strength of explicit causal integration lies in its transparency and auditability: causal constraints are interpretable, and violations can be traced to specific variables or pathways. This makes explicit approaches particularly well-suited to high-stakes settings where trade-offs between fairness, privacy, robustness, and explainability must be justified and inspected. By distinguishing spurious, as well as admissible or inadmissible causal effects, explicit models allow selective invariance-preserving task-relevant mechanisms while suppressing normatively unacceptable ones.

**Implicit causal integration.** Implicit approaches do not represent explicit causal structure, but instead encourage models to behave causally through training objectives, data diversity, sparsity, or environmental interaction.

Formally, models are trained to satisfy causal properties without explicitly encoding the causal graph $G$, but rather a set of constraints $C$, e.g., sparsity.

Implicit causal integration, by contrast, offers flexibility. It allows models to approximate causal behavior through inductive biases, multi-environment data, or counterfactual data augmentation, even when causal structure is incomplete or unknown. While they offer weaker formal guarantees, they can still soften trade-offs in practice by discouraging reliance on unstable, unethical, or spurious correlations.

**Resolving trade-offs.** From the perspective of trustworthy AI trade-offs, explicit and implicit causal integration tend to be effective in different regimes. Explicit approaches are particularly well-suited to objectives that require normative judgments about which causal pathways are admissible, such as explainable vs. proxy discrimination paths in fairness. Implicit approaches, by contrast, are often more effective for trade-offs driven by spurious correlations or distributional instability, such as robustness or privacy-utility trade-offs, especially in high-dimensional or large-scale settings where full causal structure is unavailable. In practice, many systems combine both modes, using explicit causal constraints to target specific effects while relying on implicit mechanisms to scale causal behavior more broadly.

**Integration into AI.** The necessity of causality does not depend on model scale. The key distinction lies in *how* these assumptions are integrated. In smaller or more structured ML models, causal integration is often *explicit*: causal assumptions can be embedded directly into the model class or training objective, for example, by enforcing consistency with a specified causal graph during learning (Berrevoets et al., 2024). Such explicit integration enables strong, transparent causal guarantees, but relies on the availability of reliable causal structure.

Foundation models, by contrast, rely predominantly on *implicit* causal integration. Their scale, opacity, and fixed pretraining pipelines make global explicit causal specification impractical. Instead, causal assumptions are encoded indirectly through data curation, representation learning, inductive biases, and post-training objectives that encourage invariant behavior without explicitly representing causal structure. This reliance on implicit mechanisms is not unique to foundation models. Large neural models in classical ML settings also employ sparsity constraints or counterfactual data augmentation to induce causal behavior (Burgess et al., 2018; Kim et al., 2021; Kaddour et al., 2022), but it becomes the dominant mode at scale.

Nevertheless, recent work demonstrates that explicit causal structure can still be leveraged in foundation models in a targeted or partial manner, for example, through SCM-guided entity interventions or causal regularization applied to specific components or behaviors (Wang et al., 2023; Madhavan et al., 2023). This suggests that effective causal integration in foundation models often takes a hybrid form, combining implicit causal learning at scale with localized explicit constraints where interpretability or normative guarantees are required.

## 4.2. Application to Foundational Models

For FMs, causal integration is best understood as a lifecycle process, with different leverage points at pre-training, post-training, or auditing. Importantly, these stages differ not only in when causal assumptions are introduced, but also in what causal object is being manipulated.

During *pre-training*, causal priors are introduced by intervening on the data-generating process or the representation space before task-specific objectives are defined. This includes counterfactual data generation and causal representation learning methods that aim to disentangle underlying generative factors (Rajendran et al., 2024; Jiang et al., 2024; Chen et al., 2023). At this stage, causality operates primarily at the distributional level: interventions reshape the joint data distribution so that invariant mechanisms become statistically identifiable, while spurious correlations are attenuated. Pre-training, therefore, offers a natural opportunity to encode broad structural assumptions that generalize across downstream tasks.

At the *post-training and alignment* stage, causal assumptions are enforced through fine-tuning, regularization, or alignment objectives that operate directly on model behavior (Xia et al., 2024). Rather than modifying the training distribution, constraints are expressed as requirements on how model outputs should respond to counterfactual perturbations. This makes post-training the primary locus for enforcing trustworthy AI objectives in foundation models. Fairness, or robustness constraints, can be implemented as invariance or sensitivity conditions on outputs, even when the causal structure was not explicitly modeled during pre-training.

Finally, during the *auditing and interpretability* phase, causal reasoning can be applied retrospectively, even when no causal constraints were imposed during training. Intervention-based probing, activation patching, and mechanistic interpretability methods enable estimation of the causal influence of internal components on model behavior, supporting auditing and explanation (Kissane et al., 2024; Conmy et al., 2023; Syed et al., 2024; Izadi et al., 2026). These approaches allow practitioners to assess whether learned representations and internal circuits satisfy desired invariance properties, and to diagnose failure modes related to fairness, robustness, or spurious correlations without modifying the training pipeline.

## 5. Challenges and Opportunities

Despite the advantages of a causal approach, practical applications face fundamental limitations that arise from the design choices that enable causal integration. In particular, trade-offs between explicit and implicit representations, between early and late intervention in the model lifecycle limits what guarantees can be achieved in practice. These limitations are amplified in foundation models due to their scale, opacity, and decoupled training pipelines. We outline key conceptual and practical obstacles and corresponding opportunities for navigating this design space.

### 5.1. Conceptual Challenges

**Potentially unresolvable tensions.** Not all tensions in trustworthy AI can always be fully resolved. For instance, stronger privacy protections often reduce model utility (Dwork et al., 2014; Bassily et al., 2014). We also acknowledge that some fairness conflicts stem from deeper normative or value-based disagreements; for example, a causal relationship may exist, but relying on it in decision-making may still be viewed as unfair from an ethical or legal standpoint. In these cases, causality does not eliminate trade-offs; however, it makes their structural origin explicit, shifting the debate from engineering heuristics to transparent normative choices.

**Concept superposition.** Foundation models suffer from concept superposition, namely, multiple meanings are entangled within a single representation, complicating causal reasoning (Elhage et al., 2022). This limits the granularity at which causal interventions can be applied, shifting causal control toward behavior-level constraints.

### 5.2. Implementation Challenges

**Assumptions and identifiability.** The main limitation of explicit causal integration is its reliance on accurate or partially specified causal knowledge, particularly in the form of DAGs. Misspecified graphs or incorrect structural assumptions can lead to incorrect invariances or unintended behavior. Expert-constructed DAGs may suffer from subjectivity and scalability issues, while ML-based causal discovery is constrained by identifiability assumptions and noise sensitivity. However, recent hybrid approaches combining classical causal discovery with LLM-based reasoning offer promising solutions (Afonja et al., 2024). Recent works have also shown the application of leveraging LLMs' imperfect causal knowledge to be effective (Vashishtha et al., 2023; Sheth et al., a; Hiremath et al., 2025; Sheth et al., b). In addition, approximate causal interventions are possible with a partial causal graph (Zuo et al., 2022).

Implicit approaches make weaker structural assumptions, but they are not assumption-free. They rely on properties such as environmental diversity, intervention coverage, or stability of causal mechanisms across domains. When these conditions fail, e.g., when all training environments share the same confounding structure, implicit methods may converge to spurious but stable correlations.

**Scaling.** Explicit causal approaches scale poorly with the

complexity of the causal structure, rather than with model size alone. As the number of variables, dependencies, or latent confounders grows, specifying and enforcing global causal constraints becomes difficult. In practice, explicit methods are therefore often applied locally or partially, for example, by constraining specific pathways relevant to fairness (Wang et al., 2023). Hybrid strategies that combine partial causal structure with implicit learning objectives offer a practical compromise between interpretability and scalability. Implicit approaches, by contrast, scale naturally with model and data size, as they rely on data augmentation, representation learning, and objective design rather than explicit causal structure.

**Evaluation and benchmarking.** Evaluating causal integration remains challenging due to the lack of standardized benchmarks that reflect interventional objectives. Most existing evaluations rely on observational performance, which may obscure failures under interventions. Developing benchmarks and evaluation protocols aligned with causal invariance requirements is, therefore, an opportunity for advancing trustworthy AI.

**Lack of high-quality causal data.** Applying causal integration at the pre-training phase requires high-quality interventional data, which are scarce and expensive to produce. Scalable methods for generating synthetic causal datasets show a promising direction (Webster et al., 2020; Chen et al., 2023). Alternatively, focusing on post-training methods allows causal interventions in a more data-efficient way.

**Computational complexity.** Integrating causal reasoning into large models introduces additional computational overhead, for example, through counterfactual evaluation or causal regularization during training. Parameter-efficient adaptation methods such as LoRA reduce this burden by restricting updates to low-rank subspaces, enabling efficient causal fine-tuning without modifying the full parameter set (Hu et al., 2022).

## 6. Alternative Views

Symbolic AI leverages prior knowledge through logic rules, ontologies, and formal representations, supporting structured reasoning, explainability, and generalization from small data (Díaz-Rodríguez et al., 2022). It can also help navigate trade-offs, for example, between accuracy and interpretability, by making the reasoning process more transparent and controllable. While both symbolic AI and causal reasoning rely on prior assumptions, causality uniquely enables reasoning about interventions and counterfactuals. Unlike symbolic systems that model static relationships, causal models capture how changes in one variable affect others - an essential feature in domains like fairness.

A related alternative position acknowledges the appeal of causal reasoning but emphasizes its practical limits. Locatello et al. (2019) show that disentangled representations are unidentifiable from observational data alone without inductive biases, which is consistent with our discussion in Section 4: implicit causal integration does not provide formal guarantees, but can soften trade-offs through inductive bias and environmental diversity. Rosenfeld et al. (2021) show that Invariant Risk Minimization fails to recover the optimal invariant predictor when the number of training environments is insufficient, motivating our call for diverse, multi-environment benchmarks (Section 7). Gulrajani & Hashimoto (2022) show that domain adaptation requires identifiability conditions that are often violated in practice, clarifying when causal approaches are applicable rather than dismissing them. Finally, a dominant alternative position advocates for improving model performance by scaling, and deems causal approaches as impractical. Our goal therefore, is to illustrate that implicit, explicit, or hybrid causal guidance can provide feasible, data-efficient methods for both performant and trustworthy AI models.

## 7. Conclusion and Call for Action

We demonstrate that causal models offer a principled approach to trustworthy AI by disentangling conflicting invariance requirements and shifting attention from observational to interventional accuracy. In this sense, the causal approach supports models that are both trustworthy and performant in ways that are stable under relevant interventions and justified by the underlying causal structure. We further discuss practical ways to apply causality to AI by distinguishing explicit and implicit design. To further advance the application of causality for trustworthy AI, we call for the following actions

> ✓ *Redefine trustworthy AI as a multi-objective optimization, rather than as a collection of competing constraints.* This requires establishing evaluation frameworks and benchmarks that jointly measure objectives of trustworthy AI, explicitly quantifying the trade-offs between them via multi-objective evaluation metrics.
>
> ✓ *Leverage Causality to Resolve or Soften Trade-offs*: Where possible, integrate causal reasoning to disentangle competing objectives and mitigate conflicts.
>
> ✓ *Develop Scalable Methods for Causal Data Integration*: Encourage the development of algorithms and pipelines to integrate causal knowledge into foundation models at scale.
>
> ✓ *Create and Share High-Quality Causal Datasets*: Foster initiatives to curate, annotate, and share datasets with datasets enriched with (partial) causal annotations, intervention metadata, counterfactual pairs, or multi-environment splits.

## Impact statement

This paper advances a causal perspective on trustworthy foundation models by framing fairness, privacy, robustness, and explainability as competing invariance requirements under interventions. Rather than treating trade-offs between these objectives as incidental, our analysis clarifies when they are structurally unavoidable and when they can be softened through selective causal invariance. By making the assumptions underlying these trade-offs explicit, the proposed framework supports more transparent, accountable, and principled design of learning-based AI systems, particularly in high-stakes domains such as healthcare, law, and finance.

## Acknowledgements

This work is partially funded by ELSA European Lighthouse on Secure and Safe AI funded by the European Union under grant agreement No.101070617; by the German Federal Ministry of Education and Research (BMBF): Tübingen AI Center, FKZ: 01IS18039B; by the Machine Learning Cluster of Excellence, EXC number 2064/1 Project number 390727645; and by Coefficient Giving. Views and opinions expressed are however those of the author(s) only and do not necessarily reflect those of the European Union or the European Commission. Neither the European Union nor the European Commission can be held responsible for them.

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

# A. Causal Resolution of Trustworthy-AI Trade-offs: Simulation Details

This appendix gives the technical details of the simulation referenced in section 3.5 of the main paper. Code, exact hyperparameters, and additional analyses are provided in the companion notebook https://github.com/RutaBinkyte/causal-invariance-conflicts-ai.

## A.1. Structural Causal Model

The data-generating process models 30-day clinical readmission across hospitals. The SCM DAG is provided in the main paper Figure 1.

**Variables.**

| Variable | Domain | Meaning |
|----------|--------|---------|
| $A$ | $\{0,1\}$ | Protected attribute, representing age group |
| $E$ | $\{0,1,2\}$ | Hospital environment |
| $X_1$ | $\mathbb{R}$ | Legitimate lab measurement caused by $A$ |
| $X_2$ | $\mathbb{R}$ | Spurious image-artifact intensity caused by $E$ |
| $Z$ | $\{0,1\}$ | Insurance type, treated as an unfair proxy for $A$ |
| $Y$ | $\{0,1\}$ | 30-day readmission outcome |

*Table 1.* Variables used in the simulation.

**Equations.** The structural causal model is defined as

$$
\begin{aligned}
A &\sim \text{Bernoulli}(0.5) & E &\sim \text{Uniform}\{0,1,2\} \\
X_1 &= \beta_{AX_1}A + \varepsilon_x & X_2 &= \text{artifact}(E) + \varepsilon_{\text{artifact}} \\
Z &= \mathbb{1}[\text{logit}(\beta_{AZ}A + \text{bias}_Z + \varepsilon_z) > 0.5] \\
Y &= \mathbb{1}[\text{logit}(\beta_{X_1Y}X_1 + \beta_{ZY}Z + \beta_{AY}A + \text{policy}(E) + \text{bias}_Y + \varepsilon_y) > 0.5].
\end{aligned}
$$

Parameters are set to

$$(\beta_{AX_1}, \beta_{AZ}, \beta_{AY}, \beta_{X_1Y}, \beta_{ZY}) = (0.8, 2.5, 0.2, 1.5, 1.5),$$

with

$$\text{artifact}(\cdot) = [0.0, 1.5, 3.0], \qquad \text{policy}(\cdot) = [0.2, 0.8, -0.3].$$

All exogenous noise terms are Gaussian with moderate variance.

**Causal pathways.**

| Type | Pathway | Meaning |
|------|---------|---------|
| Legitimate | $A \to X_1 \to Y$ | Admissible causal effect of $A$ |
| Legitimate | $E \to Y$ | Environment affects the outcome |
| Unfair | $A \to Z \to Y$ | Proxy-mediated effect of $A$ |
| Unfair | $A \to Y$ | Direct effect of $A$ on $Y$ |
| Spurious | $E \to X_2$ | Environment-dependent artifact with no causal edge to $Y$ |

*Table 2.* Classification of causal pathways in the example SCM.

**Splits.** We sample $N = 6000$ observations and use a stratified train/test split with a 70/30 ratio. For the robustness experiment, we define $\text{rob\_train} = \text{train} \cap \{E \neq 2\}$, and $\text{rob\_test} = \text{test} \cap \{E = 2\}$.

## A.2. Methods

All five methods (Table 3) are detailed and trained in the companion notebook: https://github.com/RutaBinkyte/causal-invariance-conflicts-ai.

| Method | Features | Intervention |
|---|---|---|
| No Intervention | $A, X_1, X_2, Z$ | None. Logistic regression on the full feature set including the protected attribute. |
| Privacy Intervention | $A, X_1, X_2, Z$ + Laplace noise | Each feature is standardized, perturbed with independent $\text{Lap}(0, 1/\varepsilon)$ noise at $\varepsilon = 1$, and unstandardized before training. |
| Fairness Intervention | $X_1, X_2, Z$ ($A$ dropped from features) | Logistic regression with a demographic-parity Lagrangian penalty $$\lambda \cdot \left| \mathbb{E}[\hat{Y} \mid A = 1] - \mathbb{E}[\hat{Y} \mid A = 0] \right|$$ added to the cross-entropy loss, at $\lambda = 1.0$. $A$ is dropped from the model's input features but is used to compute the demographic-parity term during training. |
| Partial Implicit Causal | $X_1, X_2, Z$ ($A$ dropped from features) | GroupDRO over six environments $E \in \{0, 1, 3, 4, 5, 6\}$ with varying $X_2 \mid E$, combined with counterfactual data augmentation in which $Z$ is resampled from $P(Z \mid A = 1 - A_{\text{obs}})$ holding $X_1$ and $X_2$ fixed. $A$ is dropped from the predictor's input features but is used at training time to construct the counterfactual draws. |
| Causal | $X_1, E$ | Drop $A$, $Z$, and $X_2$. Variable selection guided by the causal graph: $A$ and $Z$ are inadmissible (they carry the unfair path from the protected attribute to $Y$), $X_2$ has no causal edge to $Y$ (it is generated by $E$ but does not influence the outcome), and $X_1$ together with $E$ carries the admissible signal. |

*Table 3.* Methods compared in the simulation. The five methods are ordered along a causal-knowledge gradient: from no knowledge (No Intervention, Privacy Intervention) through partial knowledge of the sensitive variables (Fairness Intervention knows $A$ is the protected attribute; Partial Implicit Causal additionally knows $Z$ is a proxy and that environments vary across hospitals) to full knowledge of the causal graph (Causal).

## A.3. Metrics

Five trust dimensions are evaluated on the held-out test set.

**Accuracy.** Accuracy is measured as AUC-ROC on the full held-out test set, the probability that a randomly chosen positive example receives a higher predicted score than a randomly chosen negative one.

**Prediction sensitivity to $Z$.** Prediction sensitivity to the proxy variable $Z$ is measured as $\mathbb{E}_X \left[ \left| \hat{f}(X_1, X_2, Z = 1) - \hat{f}(X_1, X_2, Z = 0) \right| \right]$, computed on the test set. This is a behavioral diagnostic: it measures how much a model's prediction changes when the proxy feature is intervened on, holding the remaining observed features fixed. A lower value indicates that the prediction function is less sensitive to the unfair proxy $Z$.

**Robustness.** Robustness is evaluated using accuracy and Brier score on the held-out OOD environment $E = 2$.

The Brier score measures the mean squared error between predicted probabilities and binary outcomes:

$$\text{Brier} = \frac{1}{n} \sum_{i=1}^{n} (\hat{p}_i - y_i)^2,$$

where $\hat{p}_i \in [0, 1]$ is the predicted probability of readmission and $y_i \in \{0, 1\}$ is the observed outcome. Unlike AUC-ROC, which evaluates ranking quality, the Brier score evaluates probabilistic calibration and sharpness. A model can rank patients correctly while still assigning poorly calibrated probabilities under distribution shift; the Brier score captures this failure mode by penalizing confident but wrong predictions in the held-out hospital environment.

For binary outcomes, the Brier score lies in the interval $[0, 1]$, where lower is better. A score of $0$ corresponds to perfect probabilistic predictions, while larger values indicate worse calibrated predictions. In the radar plot (Figure 2), the OOD Brier score is converted to a higher-is-better robustness score.

**Robustness Measurement Details.** Robustness is measured on a held-out environment. We define $\text{rob\_train} = \text{train} \cap \{E \in \{0, 1\}\}$ and $\text{rob\_test} = \text{test} \cap \{E = 2\}$. Each method is retrained on samples drawn from $\text{rob\_train}$ sees only the in-distribution environments where $X_2$ has mean approximately 0.0 or 1.5. Models then evaluated on $\text{rob\_test}$, where the policy shifts and $X_2$ has mean approximately 3.0. The Partial Implicit Causal method additionally trains on synthetic hospitals $E \in \{3, 4, 5, 6\}$ as part of its multi-environment intervention; the $E = 2$ held-out evaluation applies to it equally.

**Privacy.** Privacy is measured using the adversary AUC of an attribute-inference attack on $Z$. The adversary observes the model's predictions and $X_1$, and trains its own non-DP classifier to recover $Z$ from a held-out portion of the test set. Higher adversary AUC indicates that more information about the proxy attribute $Z$ remains recoverable from the model's behavior.

**Explainability.** Explainability is measured as the fraction of test-set prediction variance attributable to causally justified features, namely $X_1$ and $E$. Operationally, we neutralize $Z$ and $X_2$ to their means, recompute predictions, and calculate the corresponding variance ratio.

**Radar-plot normalization.** For the radar plot and Pareto frontiers, all metrics are converted so that higher values are better:

$$\text{Fairness} = 1 - \frac{\text{sensitivity}}{\text{max\_sensitivity}},$$

$$\text{Robustness} = 1 - 2 \cdot \text{Brier}_{\text{OOD}},$$

and

$$\text{Privacy} = 2 - 2 \cdot \text{Adv\_AUC}.$$

## A.4. Main Results

Figure 3 reports each method's score on each of the five trust dimensions separately, with $\pm$ one-standard-deviation error bars across five seeds. Reading the chart column by column makes the per-dimension trade-offs explicit: each non-causal method scores well on at most one or two dimensions and pays on the others, while the two causal methods score high on multiple dimensions simultaneously, with the explicit Causal model winning four of five and Partial Implicit Causal winning the remaining one (Robustness).

Figure 4 projects the five-dimensional trade-off space onto three two-dimensional cross-sections: accuracy vs. fairness, accuracy vs. OOD robustness, and fairness vs. privacy. The Pareto frontiers (dashed gray lines connecting non-dominated methods) show that the causal methods are the only Pareto-optimal points on every pair, with Causal on the fairness pairs and the two causal methods sharing the accuracy-versus-OOD frontier.

**Interpretation of the Results.** The three non-causal methods each target a single trust objective and improve it at the expense of other dimensions. No Intervention prediction uses all available signal and is competitive on accuracy but worst on fairness, robustness, and privacy. Privacy Intervention adds noise to every feature indiscriminately and pays a substantial accuracy cost without fully suppressing the leakage. The correlation between the public feature $X_1$ and the sensitive attribute $A$ cam still be exploited by the adversary.

Fairness Intervention drops $A$ and adds a demographic-parity penalty ($\lambda = 1.0$); this drives the $Z$ coefficient nearly to zero and achieves a strong fairness score, but because the penalty cannot distinguish the inadmissible path $A \to Z \to Y$ from the admissible $A \to X_1 \to Y$ it attenuates both, paying a notable accuracy cost and causing the model's remaining predictive variance to migrate toward the spurious feature $X_2$, which causes a collapse on the explainability axis.

The two methods that encode causal knowledge, Partial Implicit Causal and Causal, are the only ones to achieve strong scores on several dimensions at once, but neither dominates outright. The explicit Causal model, which assumes the full graph, is best on fairness, privacy, explainability, and accuracy. These dimensions are improved by removing inadmissible and non-causal variables. The Partial Implicit Causal method, which relies only on weaker causal assumptions together with data from multiple environments, is instead best on out-of-distribution robustness, which it improves through multi-environment training rather than through the graph. This split is itself informative: the robustness advantage comes from a mechanism (environment diversity) that the single-environment graph-based model does not exploit, so the implicit method can surpass the explicit ideal on the axis where seeing many environments matters most.

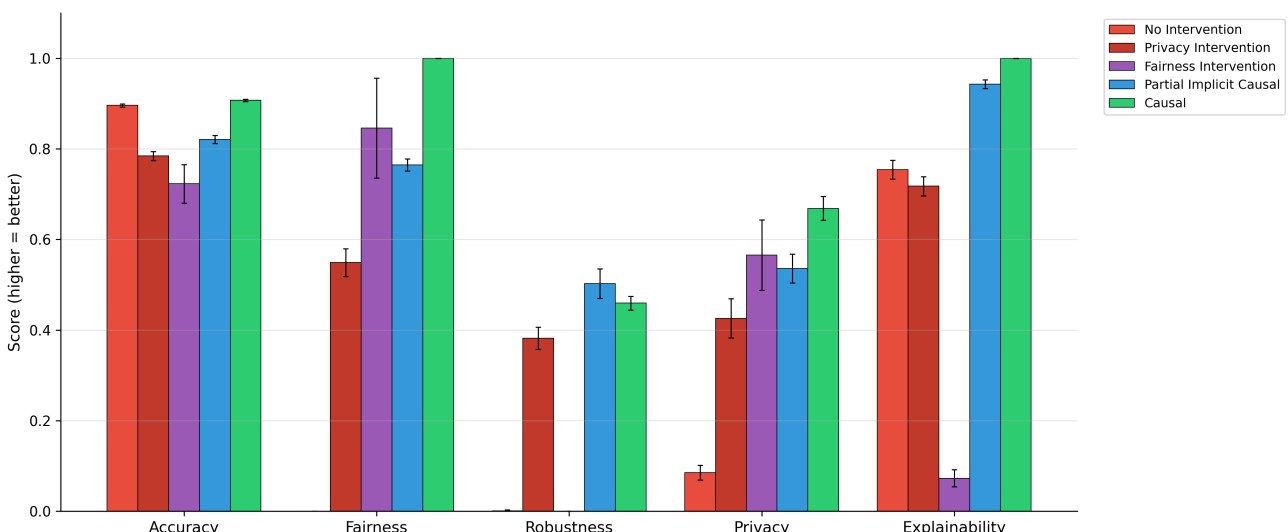

*Figure 3.* Per-dimension scores for all five methods, averaged over five seeds with ± one-standard-deviation error bars. All scores are converted so higher is better (see Appendix A.3). Each column shows one trust dimension; differences within a column compare methods on that dimension only. The Causal model is best on Accuracy, Fairness, Privacy, and Explainability; the Partial Implicit Causal method is best on Robustness.

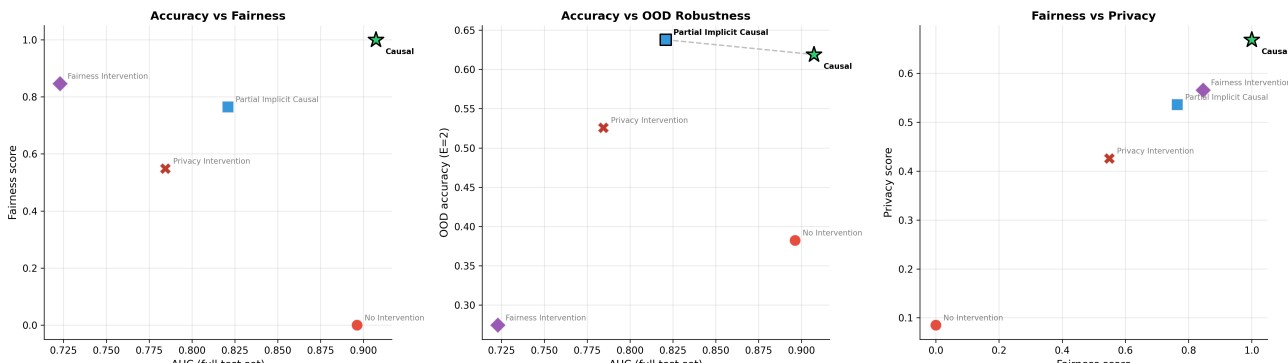

*Figure 4.* Two-dimensional Pareto projections on three contested trade-offs (multi-seed means, five seeds). A method is Pareto-optimal if no other method beats it on both axes; dashed gray lines connect the non-dominated set in each panel. Left: accuracy vs. fairness, Pareto-optimal set {Causal}. Middle: accuracy vs. OOD robustness, Pareto-optimal set {Partial Implicit Causal, Causal}. Right: fairness vs. privacy, Pareto-optimal set {Causal}.

## A.5. Limitations

**Synthetic SCM.**  The data-generating process is linear, fully specified, and known. The Causal model encodes the true graph by construction. In any real application, that graph would have to be learned, elicited, or hypothesized, with attendant error.

**Untested Generalization.**  All results come from a single SCM with one fixed set of coefficients, noise levels, and base rates, chosen to give a clear instance of the invariance conflicts the paper discusses. We did not sweep the structural parameters, vary the graph, or test nonlinear mechanisms, so we cannot claim the quantitative pattern is robust to these choices.

**Single attribute, binary proxy.**  Real fairness problems involve continuous or multi-dimensional proxies and partially identifiable structure. The $Z$-counterfactual augmentation of the Partial Implicit Causal method benefits from having a tractable distribution, $P(Z \mid A)$.

**One privacy threat model.**  The privacy metric throughout is attribute inference on $Z$. The privacy baseline included here is outperformed on this threat by methods that suppress the proxy's predictive role, because the adversary recovers $Z$ from its correlation with the public feature $X_1$. This is consistent with prior work on attribute-inference attacks (Zhao et al., 2019). The finding is threat-model-specific, not a general claim about input-noise or differentially private methods. Other threat models (membership inference, training-record reconstruction) could yield different rankings and would typically favor formal DP mechanisms.

**Explainability is causality favoring by design.**  Our explainability score is the fraction of the model's output variance that remains attributable to the causally justified features ($X_1$, and $E$) once the unfair proxy $Z$ and the spurious artifact $X_2$ are neutralized to their means. This captures one specific and narrow notion. Namely, how much of the model's behavior is driven by features we have designated as legitimate. It is favorable to the causal model by construction, since that model is built only from those features (its score is $1.0$ trivially, not as an empirical finding). A sound evaluation of explainability would require human-subject studies.

**Fairness and privacy are coupled through the same variable.**  In this SCM both threats are routed through the proxy $Z$: fairness is measured as influence through $A \to Z \to Y$, and privacy as an attribute-inference attack recovering $Z$. Any intervention that suppresses the model's use of $Z$ therefore improves both dimensions at once. This means, that fairness-only or privacy-only methods in this setting improve both, illustrating the cases, where these dimensions are in synergy, rather than the trade-off.

Crucially, however, the alignment is not merely a confound that flatters the $Z$-suppressing methods. The causal framework is what lets one know that the two objectives share a single lever: from the graph, a practitioner can read off that $Z$ both mediates the unfair path and is the attribute under privacy attack, conclude that one intervention addresses both, and act on that with justification. The non-causal baselines models obtain the joint benefit, if at all, without knowing why or being able to anticipate when it will hold.

**The Availability of $E$.**  $E$ is only available for the Causal model as an explicit feature. Other methods use $(X_1, X_2, Z)$ and receive environment information only implicitly through $X_2$, which is a noisy readout of $E$ (recall $X_2 = \text{artifact}(E) + \text{noise}$). This is consequential under distribution shift: a model relying on $X_2$ trained only on $E \in \{0, 1\}$ sees $X_2$ values from an unobserved range at $E = 2$ and extrapolates a coefficient that no longer holds, whereas the Causal model has discarded $X_2$ and instead carries $E$ explicitly. However, the inclusion of $E$ is itself a consequence of the causal analysis: the graph identifies $E \to Y$ as a legitimate pathway, so a causally-informed practitioner includes it, while a practitioner working only from the observed feature vector might not. Crucially, even with $E$ available as a feature, the Causal model still trains only on $E \in \{0, 1\}$ in the OOD experiment. The Partial Implicit Causal method, which trains across multiple $X_2 \mid E$ distributions ($E \in \{0, 1, 3, 4, 5, 6\}$), actually outperforms the Causal model in OOD tests.

## A.6. Reproducibility

The companion notebook (https://github.com/RutaBinkyte/causal-invariance-conflicts-ai) is self-contained and runs end-to-end on a standard CPU. All five seeds, sampling functions, and metric definitions are explicit in the code.

