# OpenReview forum: "Position: Trustworthy AI Suffers from Invariance Conflicts and Causality is The Solution"
_ICML.cc/2026/Position_Paper_Track — ICML 2026 Position Paper Track regular_

### Official Review · Reviewer_MNAK · 2026-03-04

**Significance:** 3
**Argument Clarity:** 3
**Rating:** 4
**Confidence:** 3

**Questions:**

1. The author notes that explicit causal methods scale poorly with the complexity of the causal structure. Could you elaborate on how a hybrid approach combining partial causal structure with implicit learning objectives  operates  during the post-training alignment phase of a foundation model?

2. In scenarios where expert-constructed DAGs suffer from subjectivity or the true causal graph is unidentifiable, how robust are the proposed implicit methods against converging on spurious but stable correlations?

**Alternative Views Section:**

Yes

**Compliance With Llm Reviewing Policy A Conservative:**

Affirmed.

**Discussion Potential:**

4

**Final Justification:**

The authors have generally addressed my concerns, I confirm my evaluation and maintain the score.

**Paper Summary:**

The paper asserts that trade-offs in trustworthy AI, such as balancing fairness, robustness, privacy, and explainability against predictive accuracy, are fundamentally caused by conflicting invariance requirements. The authors advocate for using causal reasoning as a unifying framework to resolve these conflicts, allowing models to selectively enforce invariances to suppress normatively unacceptable effects while preserving causally justified signals. They propose applying this framework across both classical machine learning and large foundation models by categorizing causal integration into explicit structural approaches and implicit inductive approaches

**Position:**

Yes

**Position In Title:**

Yes

**Related Work:**

4

**Strengths And Weaknesses:**

Strengths:
The theoretical perspective offers a unifying lens that successfully bridges classical machine learning and modern foundation models. The distinction made between explicit causal integration and implicit causal integration is valuable for scaling these concepts to large systems where specifying global causal constraints is impractical. The call to action to redefine trustworthy AI as a multi-objective optimization problem rather than a collection of competing constraints is a highly relevant and necessary direction for the research community.

Weaknesses
The paper is heavily conceptual and currently lacks empirical validation. While the hospital readmission motivating example provided in Appendix A is conceptually sound and illustrates the core problem, the paper would be significantly strengthened by moving this example to the main text and including a concrete empirical demonstration of how a causal model simultaneously improves two conflicting metrics over a standard baseline. Furthermore, the connection between causality and privacy feels less developed than the other dimensions. The proposition of targeted causal obfuscation for privacy is an intriguing design direction, but it requires more substantial evidence or a deeper mechanistic explanation to be fully convincing.

**Support:**

3

---

> ### Author Rebuttal · Authors · 2026-03-31
>
> We thank the reviewer for marking our paper for excellent discussion potential and excellent related work discussion. We address the weaknesses and answer the questions below.
>
> **1. [W1: empirical validation and examples]** We are glad, that the reviewer appreciate our motivating example and will move it to the body of the paper in the revised version. In addition we have extended it to a numerical simulation [3]. The key results are improved  OOD accuracy (0.7 vs 0.41), robustness,  bias (paths specific measure - 0 vs. 0.3), and explainability (100% vs 74% explained) as compared to statistical method. We also show improvement in privacy by removing the variables that are sensitive, but not crucial for prediction. While it is a stylized example (we will add the discussion of limitations), we believe it directly addresses the request for "a concrete empirical demonstration of how a causal model simultaneously improves two conflicting metrics" consistent with the requirements for a position paper.
>
> **2. [W2: causality and privacy]** We acknowledge the reviewers comment, that relationship between causality and privacy has less support from the literature. Indeed, causality and privacy is  the least explored area in the current literature, however pioneering work, such as Tople et al. shows promising direction. The goal of this paper is to further encourage this direction and suggest ideas. We hope we start a discussion which will lead to concrete evidence and examples.
>
>
> **3. [Q1: hybrid post training alignment ]** We thank the reviewer for the interesting suggestion that builds upon our definition of explicit and implicit causality. We foresee that this could be done in post-training alignment phase where partial causal structure is used to define targeted interventions, while implicit learning objectives enforce the desired behavior at scale. Concretely, a limited amount of causal knowledge such as sensitive attributes or specific causal pathways is specified explicitly. For example, via a partial graph or intervention rules. Importantly, this explicit structure is not embedded as a full causal model, but is instead used to determine where invariance or sensitivity constraints should apply. The model itself remains implicit: it is trained using counterfactual augmentation or multi-environments rather than through an explicit structural representation.
>
> **4. [Q2: spurious stable correlations]** We agree that when expert-specified DAGs are subjective or the true causal structure is unidentifiable, both explicit and implicit approaches face fundamental limitations. This is a known limitation and is consistent with our discussion in Section 4.1, where we emphasize that implicit methods provide weaker guarantees compared to explicit causal modeling. Our contribution is not to claim that implicit methods resolve this problem, but rather to clarify when and why such failures occur. In our framework, these failures arise precisely because the invariance constraints induced by the training setup do not align with the true causal invariances of the data-generating process. We agree that reducing subjectivity in causal assumptions is an important direction. Recent work suggests that aggregating imperfect or partially reliable experts, potentially including LLM-based reasoning can improve causal discovery under uncertainty [1,2]. We view such approaches as complementary: they can help refine or approximate causal structure, which can then be used either explicitly (as constraints) or implicitly (via better-designed environments or interventions).
>
> We thank reviewer for their thoughtful comments and actionable suggestions for improving the paper and are happy to answer additional questions.
>
> **References and Experimental Details**
>
> [1]Causal Order: The Key to Leveraging Imperfect Experts in Causal Inference
>
> [2]From Guess2Graph: When and How Can Unreliable Experts Safely Boost Causal Discovery in Finite Samples?
>
> [3] Simulation Details: We define a structural causal model (SCM):
>
> $A \sim \mathrm{Bernoulli}(0.5), \quad E \sim \mathrm{Categorical}(\{0,1,2\})$
>
> $X_1 = \beta_{AX} \cdot A + \varepsilon_{X_1} \quad \text{(causal feature; e.g., lab result)}$
>
> $X_2 = \mathrm{artifact}(E) + \varepsilon_{X_2} \quad \text{(spurious feature; measurement artifact)}$
>
> $Z = \mathbf{1}\\left[\sigma\\left(\beta_{AZ} \cdot A + b_Z + \varepsilon_Z\right) > 0.5\right] \quad \text{(proxy variable; e.g., insurance status)}$
>
> $Y = \mathbf{1}\\left[\sigma\\left(\beta_{X_1 Y} \cdot X_1 + \beta_{ZY} \cdot Z + \beta_{AY} \cdot A + \mathrm{policy}(E) + b_Y + \varepsilon_Y\right) > 0.5\right]$
>
> **Metrics used in the simulation:**
>
> *Accuracy* — AUC-ROC on a held-out test set.
>
> *Fairness* — Path-specific unfairness through Z.
>
> *Robustness* — Accuracy + Brier score under domain shift.
>
> *Privacy* — Attribute inference attack on Z.
>
> *Explainability* — Fraction of prediction variance from causal features.

---

> > ### Author Rebuttal · Reviewer_MNAK · 2026-04-01
> >
> > I thank the authors for the resonses. The simulation results are promising, but the summary lacks details that would help assess credibility. I have few follow-up questions. Could the authors clarify:
> >
> > - What are the exact dataset sizes, train/test/environment splits?
> >
> > - Are the differences statistically significant (e.g., 95% CI or p-values)?
> >
> > - How precisely are the metrics computed? What is the formula or algorithmic procedure for computing "Explainability" and how "Path-specific unfairness" is this estimated?

---

### Official Review · Reviewer_TF8x · 2026-03-09

**Significance:** 2
**Argument Clarity:** 2
**Rating:** 3
**Confidence:** 4

**Questions:**

Please refer to the Strengths and Weaknesses section.

**Alternative Views Section:**

Yes

**Compliance With Llm Reviewing Policy A Conservative:**

Affirmed.

**Discussion Potential:**

2

**Final Justification:**

I thank the authors for responding to my comment. I am aware that the authors mention implicit / incomplete causal information in several places. My only concern was that they do not provide any concrete recipe for how to go about realizing their proposal in such scenarios. While they make some references in their follow-up, i.e., "partial causal annotations, intervention metadata, counterfactual pairs, or multi-environment data", it is not clear how to leverage these to (i) measure invariance conflicts and then, (ii) to understand trustworthy AI trade-offs. While I agree that it is important to frame trustworthy AI as a multi-faceted concept with several conflicting conditions, under what circumstances such conflicts could be identified in a clear way in practice (and possibly, how so) remains still to be understood. It is due to this lack of clarity that I would maintain my current score.

**Paper Summary:**

The authors advocate for the usage of causal knowledge when optimizing trustworthiness objectives in machine learning models. They claim that when the causal graph corresponding to the data generating process is considered, different trustworthiness objectives could be seen as being in conflict with one another. Benchmarks and evaluations should then be designed accordingly to account for such trade-offs in a systematized manner.

**Position:**

Yes

**Position In Title:**

Yes

**Related Work:**

2

**Strengths And Weaknesses:**

## Strengths:

1. The authors highlight the need to shift the optimization target in current machine learning from performance under the observational distribution to correctness under interventions, which is quite important. Conventional machine learning research, especially in the recent times, often stop at claiming superior accuracy on some set of benchmarks over existing works but rarely investigate whether the attained performance gains come from learning true causal structures in the data or merely through overfitting - especially as dataset sizes continue to grow. This approach continues to maintain the black-box nature of the these models (large models in particular), if not exacerbate the issue.


2. The authors unify four of the central pillars of trustworthy AI - privacy, fairness, robustness, and explainability - as mechanisms and constraints derived from the true causal graph governing the data generating process. This unified view allows for a systematic study of trustworthy AI, which not only provides a unified view clarifying interactions between these pillars, but also simultaneously benefitting from the tools developed in the causality community.



## Weaknesses:


1. While quite appealing, the whole set of propositions made in this paper assume and rely on the availability of the causal graph describing the true data generating process. This is one of the hardest known problems in machine learning, often provably impossible in many practical scenarios [a]. Therefore, requiring access to something as powerful as the causal graph seems too idealized and strict of a condition, the possibility of which actually being realized remaining rather low based on currently available capabilities.

2. It is not clear how each of the trustworthiness criteria could be derived and their interactions modeled via a causal graph in practice. The authors provide example cases of causal resolution for each of the trustworthiness criteria and how trade-offs can occur between them but no concrete framework / proposal for how they could be unified under a single umbrella and studied in a systematized manner in the language of causality. This lack of clarity leaves the position in a vague state where the idea sounds sensible only superficially while being devoid of any clear path or initial steps towards realization. Additionally, no concrete example has been illustrated where all four trustworthiness metrics could be tested simultaneously and one that shows the said trade-offs in action in an actual causal graph - something that is imperative for a position of this sort.

3. The Alternative Views include those of Symbolic AI and Scaling, both of which are unrelated to at least one of the trustworthiness objectives (e.g., Symbolic AI is unrelated to robustness, Scaling is unrelated to explainability, etc.), in which case, they risk tending towards strawman arguments. The authors should spend more time thinking about more related, competing alternative views, such as those actually from the causality literature that illustrate its practical limits [a, b, c] (but they need not be limited in this regard - all that matters is that the views remain related to the entirety of the scope of the paper).



### References:

[a] Locatello et al., "Challenging common assumptions in the unsupervised learning of disentangled representations", ICML 2019.\
[b] Rosenfeld et al., "The Risks of Invariant Risk Minimization", ICLR 2021.\
[c] Gulrajani and Hashimoto, "Identifiability Conditions for Domain Adaptation", ICML 2022.

**Support:**

2

---

> ### Author Rebuttal · Authors · 2026-03-31
>
> We are happy the reviewer appreciates our call for unification of  trustworthy AI objectives and “need to shift the optimization target in current machine learning from performance under the observational distribution to correctness under intervention”. We respond to the reviewers questions below:
>
> **1. [W1: causal graph]** We thank the reviewer for this concern, that is common in the community, and is one of the point our position paper aims to address. Namely, our paper is the first to explicitly distinguish explicit causal integration (requiring causal graph) and implicit (does nor require fully specified graph) in Section 4.1. Implicit approaches, such as invariant feature learning, counterfactual data augmentation and other, achieve causality through inductive biases and multienvironment learning rather a DAG. In Section 5.2 we discuss how causality is useful even when the graphs are unavailable or partial. We believe the distinction of explicit and implicit causality or the intermediate variants is crucial for forwarding the discussion on the use of causality for ML and FMs, and will make this distinction more clear in the introduction.
>
> **2.[W2: no unified framework]** The reviewer acknowledges the importance of the unified framework (also our position). We would like to draw the reviewers attention to  the conflicting invariance lens which we propose is exactly the first unified for trustworthy AI trade-offs and provides a common vocabulary for fairness, privacy, robustness, explainability and causal resolutions. We argue that this unifying lens is the core contribution of our position paper, that opens possibility for a interdisciplinary discussion on dimensions of trustworthy AI and causality, that previously lacked the common language.
>
> **3.[W3: concrete examples]** We point the reviewer to the examples from the literature (eg. Tople et al Chiappa et al) discussed in the paper. In addition, we complement our illustrative example (Figure 1, Appendix A) with a numerical simulation illustrating how a causal model can improve results across all dimensions compared to the statistical model (see the answer to **Reviewer VaYA**), and as per request by **Reviewer MNAK**, we will include it in the revised version of the paper.
>
> **4. [W4: alternative views]** We thank the reviewer for valuable suggestions and will incorporate all 3 papers into the alternative views section. We will add the following discussion. *Locatello et al, 2019* show that unidentifiable from the observations alone without inductive biases. This is consistent with and will be added to  Section 5.2, where we acknowledge that implicit causality, does not provide formal guarantees, but can soften the trade-offs. *Rosenfeld et al., 2021*, show, that IRM fails to recover the optimal invariant predictor if the number of environments is insufficient. We will add it to the alternative views, and as a motivation for call for action for diverse environment benchmarks. *Gulrajani and Hashimoto 2022*, show that domain adaptation requires identifiability conditions that are often violated. We will emphasize it in the discussion of when the causal approaches are applicable. Finally, we acknowledge that symbolic approach does not cover robustness and will make it explicit in the revised version.
>
> We thank the reviewer for their time and suggestions for improving the paper and remain available to provide additional clarifications.

---

> > ### Author Rebuttal · Reviewer_TF8x · 2026-04-02
> >
> > I thank the authors for taking the time to respond to my comments. While most of them have been addressed, I still have one remaining concern: regarding the concrete example, although the authors illustrate how the causal graph can unify trustworthiness criteria under a single umbrella through the lens of conflicting invariances, it still relies on the strong assumption of the causal graph being available. While they mention practical scenarios where the causal graph is implicit and as a result, not fully specified, their example does not take this into account. The reason I am particularly concerned about this is due to the feasibility of realizing the propositions made - most tasks that are performed via implicit causal integration (which constitute the majority of real-world tasks) are done that way because obtaining the causal graph is prohibitively difficult. If such is the case, then the suggested calls for action such as redefining trustworthy AI objectives, curating causal datasets, etc., are impractical.

---

### Official Review · Reviewer_4nw2 · 2026-03-11

**Significance:** 3
**Argument Clarity:** 3
**Rating:** 4
**Confidence:** 3

**Questions:**

See weaknesses.

**Alternative Views Section:**

Yes

**Compliance With Llm Reviewing Policy A Conservative:**

Affirmed.

**Discussion Potential:**

3

**Final Justification:**

As discussed during the rebuttal period, the paper is clearly written and describes how a causal lens can benefit a number of areas in trustworthy machine learning. My main concern, shared by other reviewers, is that the lack of concrete solution concepts for how to *practically* apply causal assumptions to trustworthy machine learning may limit the degree to which this paper would spur discussion, and thus mark it as a borderline accept.

**Paper Summary:**

Research into particular goals of trustworthy AI, including fairness criteria, robustness, and privacy, has established fundamental impossibility results on either simultaneously optimizing multiple of these goals, or seeking to optimize for trustworthy qualities while not decreasing accuracy. The authors argue that revisiting these goals with a causal perspective can help soften or even fully dismantle these impossibilities, as well as understand why they occur in the first place. In particular, each property can be viewed as wanting a particular characteristic to remain invariant under certain interventions.

**Position:**

Yes

**Position In Title:**

Yes

**Related Work:**

3

**Strengths And Weaknesses:**

**Strengths**
- I found the argument interesting, and was convinced that bringing in causal assumptions could provide ways to more deeply understand and adjust existing definitions of trustworthy AI.
- The authors do a thorough job of engaging with limitations, such as discussing the practical limitations of causal methods.

**Weaknesses**
- The introduction and other discussion mentions foundation models, LLMs, etc. However, the preliminaries were stated in a more standard learning setting with inputs X and labels Y. I would have liked some discussion on how these definitions can be translated to the LLM setting as claimed, as arguably even the notions of trustworthiness such as fairness become much more complicated and subtle than just a demographic parity condition.
- Perhaps this is more due to my unfamiliarity with causality, but it seemed like there was some ambiguity in terms of how interventions and invariance are defined. For instance, it seems like for something like robustness or fairness, the intervention is some sort of change applied to the test distribution, and the invariance is measured by some metric of the original model on the train vs test distribution. On the other hand, for privacy, it seemed like the intervention was used to create a different training distribution, and the invariance was measured in terms of the difference between a model trained on either of the two distributions. It would be useful to have some discussion of these differences, and why they make sense to be presented together, as they seem quite different to me.
- I would have liked some discussion on the potential harms/benefits of the necessary assumptions that need to be made in any sort of causal approach. It seems like much of the trustworthy AI community values guarantees that can be achieved via minimal assumptions, as each additional assumption is a way that the promised trust can fail if the assumption is violated. For instance, differential privacy definitions are specifically designed to make no assumption about the underlying data distribution in order to provide truly worst-case guarantees. I think from this point of view, building a private system that relies on a number of causal assumptions about how the data is generated might be a serious privacy concern. While I agree that adding some of these assumptions can make some of these goals more tractable to achieve, it would be useful to add a discussion on when when the benefit of adding these assumptions is worth the harm that might arise if some of the assumptions are incorrect.

**Support:**

3

---

> ### Author Rebuttal · Authors · 2026-03-31
>
> We thank the reviewer for finding the argument in interesting, the discussion of limitations thorough, and are delighted, that the reviewer was “convinced that bringing in causal assumptions could provide ways to more deeply understand and adjust existing definitions of trustworthy AI.” We answer the reviewers questions below.
>
> **1. [W1: FM preliminaries]** We thank the reviewer for the important note. In LLMs the fairness invariance requirement can be formalized as follows. The X denotes the prompt or context and Y corresponds to the completion. In this setting, the fairness invariance requirement means, that given a sensitive attribute A (eg. gender) the change in X from “he” to “she” should not change the output Y, eg. recommendation letter content. We will clarify this in the revised version.
>
> **2.[W2: Ambiguity in invariance definition]** We once again thank the reviewer, for pointing out that privacy uses training level invariance, while robustness or fairness uses test level invariance. This observation adds to strengthening the unifying invariance framework and explaining the conflicts between different requirements at different stages. We will add clarification table on the details and stages of each invariance requirement to the revised version.
>
> **3. [W3: impact of incorrect causal assumptions]** We thank the reviewer for raising an important point. We note, that while statistical models have implicit assumption, causal models rely on explicit assumptions, for which  impact can be formally bounded using sensitivity analysis. Regarding the DP, it is correct, that misspecified causal privacy model may leak sensitive information. However, in practice, DP has a strong impact on accuracy, therefore is often replaced by pseudo anonymization (eg. removing the names) that does not provide any formal guarantees. In this case, even an imperfect causal model that reduces leakage at acceptable accuracy is preferred. We will add this clarification to the discussion.
>
> We thank again the reviewer for thoughtful comments and useful suggestions and are happy to answer any additional questions.

---

> > ### Author Rebuttal · Reviewer_4nw2 · 2026-04-04
> >
> > Thank you to the authors for addressing my review. I will maintain my score.

---

### Official Review · Reviewer_VAYa · 2026-03-13

**Significance:** 3
**Argument Clarity:** 4
**Rating:** 4
**Confidence:** 3

**Questions:**

Q: The paper argues that causality can play an important role in understanding and resolving conflicts among trustworthy AI objectives. While this position paper uses a causal-inference framework to *explain* these trade-offs, concrete cases where causal approaches have actually *resolved or meaningfully mitigated* them in real systems appear comparatively limited.

In this context, could the authors point to specific examples, ideally discussed more explicitly in the paper, where a causal approach has led to tangible improvements in such trade-offs in practice? Grounding the position in such evidence would substantially strengthen the case for causal learning as an actionable framework, rather than primarily a descriptive one.

(Minor) Typo: L334 - modelslarge

**Alternative Views Section:**

Yes

**Compliance With Llm Reviewing Policy A Conservative:**

Affirmed.

**Discussion Potential:**

3

**Paper Summary:**

**TL;DR**

The requirements of trustworthy AI, e.g., fairness, privacy, robustness, explainability, and accuracy/utility, can each be interpreted as a demand for the model to remain invariant under specific types of change. However, these invariance requirements frequently conflict with one another. The authors term these conflicts *invariance conflicts*, and argue that resolving them requires consideration of causal structure to determine which changes should be blocked and which should be permitted.

---

**Core Idea 1: Trustworthy AI Objectives as Invariance Requirements**

The paper reframes various trustworthy AI objectives as invariance demands with respect to specific perturbations:

- Fairness: Model decisions should not change substantially when protected attributes are varied
- Privacy: Whether or not a specific individual's data is included should not significantly affect model outputs
- Robustness: Model performance should remain stable under environmental changes or distribution shifts
- Explainability: The model should be unresponsive to meaningless changes, while responding consistently to meaningful ones

In short, trustworthy AI objectives can be understood as a collection of distinct and heterogeneous invariance requirements.

---

**Core Idea 2: Trade-offs as Conflicts Between Invariance Requirements**

Accuracy seeks to exploit any correlation useful for prediction, while fairness and robustness restrict the use of specific causal pathways or spurious features. Trade-offs arise precisely because different objectives impose incompatible invariance conditions on the same model.

---

**Core Idea 3: Causality as a Tool for Resolving These Conflicts**

Simultaneously satisfying multiple invariance conditions requires distinguishing spurious correlations from stable mechanisms - a task that cannot be reliably accomplished from observational data alone - and therefore necessitates causal reasoning.

The authors particularly emphasize selective invariance: rather than eliminating all dependencies, the goal is to block specific causal paths while preserving others that are legitimate.

---

**Proposed Directions**

To operationalize this, the paper proposes two complementary approaches:

- Explicit causal modeling: Directly representing dependency structures through causal graphs or structural causal models (SCMs)
- Implicit causal integration: Indirectly instilling causal properties into foundation models through data design, training objectives, or architectural choices

**Position:**

Yes

**Position In Title:**

Yes

**Related Work:**

4

**Strengths And Weaknesses:**

**Strengths**

- Clarity of Position:

The paper presents a clear and well-defined position: that causal reasoning can serve as a unifying framework for understanding and resolving conflicts between trustworthy AI objectives. The framing of fairness, privacy, robustness, and explainability as invariance requirements is a conceptually interesting perspective that adds clarity to an otherwise fragmented landscape.

- Support of Argument:

The paper reinterprets trade-offs among trustworthy AI objectives as invariance conflicts, and proposes selective invariance, leveraging causal reasoning to selectively preserve or block specific causal pathways, as a principled resolution strategy. While empirical validation is limited, the argument is presented through a coherent and internally consistent conceptual structure that makes the position reasonably persuasive.

- Relevance and Importance:

Trustworthy AI is one of the most actively pursued research themes in the current ML community. The paper's attempt to offer a causality-grounded, integrative framing of this space represents a meaningful contribution as a position paper.

---

**Weaknesses**

- Capacity to Stimulate Discussion:

The paper introduces a novel framing of the relationship between trustworthy AI and causality, and has the potential to prompt some degree of community discussion. However, it is unclear whether this perspective can substantially advance the conversation beyond what already exists. Causal ML is conceptually attractive, but faces well-documented challenges in practice, and there is a real risk that discussion provoked by this paper will ultimately circle back to the familiar limitations of causal ML, rather than opening genuinely new directions.

- Ambiguity of Actionable Directions:

The paper proposes several directions, e.g., including explicit SCM-based modeling and implicit causal learning, but leaves largely open the question of how these approaches would concretely resolve trustworthy AI trade-offs in practice. Fundamental challenges inherent to causal ML, including identifiability, scalability, and computational complexity, are acknowledged but not adequately addressed. As a result, the proposed directions remain aspirational rather than actionable.

**Support:**

3

---

> ### Author Rebuttal · Authors · 2026-03-31
>
> We are happy the reviewer find our position “clear and well-defined” as well as praise for support of the argument and relevance and importance of the topic.To address the weaknesses and questions:
>
> **1.  [W1: limited discussion potential]** We appreciate the concern and argue, that providing a unified language for the trustworthy AI -    invariance requirements - provides ground for the discussion in the community that currently lacks unified vocabulary. In addition, distinguishing implicit and explicit causality allows discussion on the alternative approaches and expanded boundaries of causality.
>
> **2.[W2,Q1: specific examples how causality can solve trade-offs]** We point out to two examples from the literature, discussed in the paper and our illustrative example, for which we implemented a numerical simulation (to be added to the revised version).
>
> *A) Chiappa (2019, path-specific fairness)* shows how blocking only the discriminatory causal path preserves close to   baseline accuracy while satisfying fairness.
>
>  *B) Tople et al. (2020)* show causal models reduce membership inference attack success while requiring a  smaller DP budget ε, simultaneously improving privacy and utility.
>
> *C) In our numerical simulation* (Details in [1]) based on the example in Figure 1 and Appendix A we improve we compare causal method (full graph) to statistical method and show how OOD accuracy (0.7 vs 0.41), robustness,  bias (paths specific measure - 0 vs. 0.3), and explainability (100% vs 74% explained) is improved as compared to statistical method. We also show improvement in privacy by removing the variables that are sensitive, but not crucial for prediction. We acknowledge that this idealized example and discuss the limitation, however, we believe the example serves well for starting a discussion and focusing future research efforts. The plots of the results are available at this anonymized link: https://osf.io/wxg83/overview?view_only=51c07a68e88e4a7885730eab6d84e6a9
>
> **3. [W2: limitations and actionable directions]** We note, that in  Section 5.2, we also provide actionable existing solutions for identifiability, scalability, and computational complexity, such as hybrid approaches, partial graphs, LLM-assisted causal discovery. In addition, we call for actions such as causal data sets and benchmarks, and scalable causal knowledge integration approaches. We do not claim, that these limitations are solved, but we argue that they are tractable and point to the directions based on existing successful implementations.
>
> We thank the reviewer for a thougtful review and remain available for additional questions or clarifications.
>
>
> **References and Experiment Details**
>
>  [1]   Simulation Details: We define a structural causal model (SCM):
>
> $A \sim \mathrm{Bernoulli}(0.5), \quad E \sim \mathrm{Categorical}(\{0,1,2\})$
>
> $X_1 = \beta_{AX} \cdot A + \varepsilon_{X_1} \quad \text{(causal feature; e.g., lab result)}$
>
> $X_2 = \mathrm{artifact}(E) + \varepsilon_{X_2} \quad \text{(spurious feature; measurement artifact)}$
>
> $Z = \mathbf{1}\\left[\sigma\\left(\beta_{AZ} \cdot A + b_Z + \varepsilon_Z\right) > 0.5\right] \quad \text{(proxy variable; e.g., insurance status)}$
>
> $Y = \mathbf{1}\\left[\sigma\\left(\beta_{X_1 Y} \cdot X_1 + \beta_{ZY} \cdot Z + \beta_{AY} \cdot A + \mathrm{policy}(E) + b_Y + \varepsilon_Y\right) > 0.5\right]$ (readmission risk)
>
> **Metrics used in the simulation:**
>
> *Accuracy* — AUC-ROC on a held-out test set.
>
> *Fairness* — Path-specific unfairness through Z.
>
> *Robustness* — Accuracy + Brier score under domain shift.
>
> *Privacy* — Attribute inference attack on Z.
>
> *Explainability* — Fraction of prediction variance from causal features.

---

> > ### Author Rebuttal · Reviewer_VAYa · 2026-04-03
> >
> > Thank you for the rebuttal, I appreciate the clarifications and additional references.
> >
> > > [W1: Limited discussion potential]
> >
> > I agree that the proposed unified perspective is valuable. Framing objectives through invariance requirements provides a useful common language, and I can see how this may facilitate more structured discussions within the community. That said, I am still uncertain whether this alone is sufficient to justify a strong contribution as a position paper.
> >
> > > [W2: Specific examples]
> >
> > Thank you for sharing the references. For the final version, it would be helpful if one of these literature (not numerical experiment) were discussed in more concrete detail. While the current manuscript does mention prior work where causal inference has been used to address such trade-offs, the presentation remains somewhat high-level. As a result, it felt more like “such work exists” rather than providing a clear understanding of how these approaches concretely resolve the trade-offs. A more detailed walkthrough of at least one example could make the argument more compelling and better support the goal of stimulating discussion.
> >
> > Overall, the rebuttal has made me more positive about the paper. However, I do not feel strongly enough to advocate for acceptance, and will therefore keep my current score.

---

### Decision · Program_Chairs · 2026-04-30

**Decision:**

Accept (regular)

**Comment:**

Trustworthy AI is an important topic and this work offers an interesting proposition: many of the challenges in the field can be solved with the use of causal structure. This is an interesting proposition, and most reviewers found it worthy of publication as a position paper. However, there was also a discussion about the feasibility of this approach. The benefits of causal modeling are known and leading researchers have advocated for adopting them [1,2]. But inferring causality is challenging from observational data [3]. The authors claim that their model does not assume full causal graph (DAG).

Therefore, this paper could be very beneficial if its recommendations can be realized. However, it is not clear if this is indeed the case.


[1] Pearl, J. (2018). Theoretical impediments to machine learning with seven sparks from the causal revolution. arXiv preprint arXiv:1801.04016.

[2] Schölkopf, B. (2022). Causality for machine learning. In Probabilistic and causal inference: The works of Judea Pearl (pp. 765-804).

[3] Liu, T., Ungar, L., & Kording, K. (2021). Quantifying causality in data science with quasi-experiments. Nature computational science, 1(1), 24-32.